# Astronomical component estimation (ACE v.1) by time-variant sinusoidal modeling.

Matthias Sinnesael[1], Miroslav Zivanovic[2], David De Vleeschouwer[1,3], Philippe Claeys[1], Johan Schoukens[4]

[1] Analytical, Environmental, & Geo-Chemistry, Vrije Universiteit Brussel, 1050 Brussels, Belgium.
[2] Department of Electrical and Electronic Engineering, Universidad Pública de Navarra, 31006 Pamplona, Spain.
[3] MARUM, Center for Marine Environmental Science, Leobener Strasse, 28359 Bremen, Germany.
[4] Department of Fundamental Electricity and Instrumentation, Vrije Universiteit Brussel, 1050 Brussels, Belgium.

*Correspondence to*: Matthias Sinnesael (masinnes@vub.ac.be)

**Abstract:**

Accurately deciphering periodic variations in paleoclimate proxy signals is essential for cyclostratigraphy. Classical spectral analysis often relies on methods based on (Fast) Fourier Transformation. This technique has no unique solution separating variations in amplitude and frequency. This characteristic can make it difficult to correctly interpret a proxy's power spectrum or to accurately evaluate simultaneous changes in amplitude and frequency in evolutionary analyses. This drawback is circumvented by using a polynomial approach to estimate instantaneous amplitude and frequency in orbital components. This approach was proven useful to characterize audio signals (music and speech), which are non-stationary in nature. Paleoclimate proxy signals and audio signals share similar dynamics; the only difference is the frequency relationship between the different components. A harmonic frequency relationship exists in audio signals, whereas this relation is non-harmonic in paleoclimate signals. However, this difference is irrelevant for the problem of separating simultaneous changes in amplitude and frequency.

Using an approach with overlapping analysis frames, the model (Astronomical Component Estimation, version 1: ACE v.1) captures time variations of an orbital component by modulating a stationary sinusoid centered at its mean frequency, with a single polynomial. Hence, the parameters that determine the model are the mean frequency of the orbital component and the polynomial coefficients. The first parameter depends on geologic interpretations, whereas the latter are estimated by means of linear least-squares. As output, the model provides the orbital component waveform, either in the depth or time domain. Uncertainty analyses of the model estimates are performed using Monte Carlo simulations. Furthermore, it allows for a unique decomposition of the signal into its instantaneous amplitude and frequency. Frequency modulation patterns reconstruct changes in accumulation rate, whereas amplitude modulation identifies eccentricity-modulated precession. The functioning of the time-variant sinusoidal model is illustrated and validated using a synthetic insolation signal. The new modeling approach is tested on two case studies: (1) a Plio-Pleistocene benthic $\delta^{18}O$ record from ODP Site 846 and (2) a Danian magnetic susceptibility record from the Contessa Highway section, Gubbio, Italy.

## 1 Introduction

Variations in solar radiation received by the Earth are caused by quasi-periodic changes in its astronomical parameters: precession, obliquity and eccentricity. These quasi-periodic oscillations induce climate variations, which are often recorded in sedimentological archives (Hinnov, 2013). The periodicities of the different orbital cycles are reasonably well constrained for the last 50 Ma (Berger et al., 1992; Laskar et al., 2004; 2011a, b; Westerhold et al., 2012; Waltham, 2015) and over the last 20 years, the application of astrochronology and cyclostratigraphy to numerous records throughout the Cenozoic led to significant improvements of the Geological Time Scale (both in precision and accuracy, Hinnov and Hilgen, 2012 *in* Gradstein et al., 2012). Moreover, the detection of an astronomical imprint in sedimentological archives sheds light on the relative contribution of orbital parameters to climate variability, and provides hints to which parts of the globe (tropical vs. high latitudes) dominantly drive global climate change.

The identification and extraction of the orbital components is essential in studies applying the astronomical theory (Muller and MacDonald, 2000). The interpretation of those components converts distance-series into time-series. Various techniques for distance- and time-series analysis allow for this quantitative assessment in a sedimentological archive. Spectral analysis constitutes the foundation of this process (Hinnov, 2013). A broad range of spectral analysis techniques exists. A periodogram depicts the spectral power of a signal using discrete (Fast) Fourier Transformations (FFT). Thompson (1982) introduced the multi-taper method (MTM) to overcome some of the limitations of conventional Fourier Transformations. The MTM harmonic analyses are also applied on moving analysis frames (i.e. moving window approach) with the aim to localize principal orbital components jointly in the time and frequency domain, for example through evolutive harmonic analysis (Meyers et al., 2001; Pälike et al., 2006; Meyers and Hinnov, 2010). In this study the term analysis frame is preferred over the term (moving window) to avoid confusion with the window (or tapering) function terminology which is often encountered in signal processing too. Different astrochronologic studies rather use moving analysis frame FFT (e.g. Lourens and Hilgen, 1997; Martinez et al., 2012, 2015; Yao et al., 2015) or continuous wavelet transform, describing the signal through a scalogram – a bank of user-defined filters with time-varying non-uniform resolution (e.g. Torrence and Campo, 1998; De Vleeschouwer et al., 2014).

The aforementioned methods and most other commonly used in cyclostratigraphy are non-parametric in nature, which means that no assumption about the origin of the data is taken into account. Accordingly, they cannot ensure unique instantaneous amplitude-frequency description of a signal (Boashash, 1992) and therefore can hamper a correct interpretation of a proxy series, as discussed for example in Meyers et al. (2001) or Laurin et al. (2016). Hiatuses, noise and changes in sedimentation rate are additional sources of uncertainty (e.g. Meyers and Sagemann, 2004; Meyers et al., 2008).

Recently, a parametric method a.k.a. TimeOpt model has been introduced (Meyers, 2015) that combines band-pass filtering, the Hilbert transform and specific signal modeling to describe the precession-band envelope from stratigraphic data. The signal model consists of a set of sinusoids with constant amplitudes plus noise. The model parameter (amplitudes) estimates are obtained by linear least-squares regression across a fine grid of sedimentation rates and the one providing the

largest goodness-of-fit is retained. A constraint of this approach is related to the assumption that sedimentation rate is constant within the analysis frame – stratigraphic interval analyzed. For certain cases this might be approximately true; in a more general scenario, however, such an assumption will give rise to model-data mismatch.

This study circumvents this drawback by using a polynomial modeling approach to estimate and extract instantaneous amplitude and frequency in orbital components. The approach has been proven useful to characterize audio signals (music and speech), which are non-stationary in nature (Zivanovic and Schoukens, 2010; 2012). Paleoclimate proxy signals and audio signals share similar dynamics; the only difference is the frequency relationship between the signal components. A harmonic frequency relationship exists in audio signals, whereas this relation is non-harmonic in paleoclimate signals. By dropping the harmonicity constraint in the model for audio signals, a proxy signal is conceived as a collection of non-harmonic sinusoids whose instantaneous amplitude and frequency vary over the record. Those variations are captured by polynomials whose coefficients describe the relationship between varying amplitude and frequency of the sinusoids in an analysis frame. The main benefit of this approach is that short-term variations in sedimentation rate can be captured by the proposed method, as the signal model allows for instantaneous frequency change in the analysis frame. Moreover, for each data sample the sedimentation rate can be estimated leading to the concept of instantaneous sedimentation rate. The signal model identification (parameter estimation) boils down to solving a system of linear equations. Additionally, a measure for the uncertainty of the model estimates is given by means of Monte Carlo simulations.

After introducing the reader to the ACE v.1 model, distance-series analysis on a modified insolation data for the last six million years as modeled by Laskar et al. (2004) is carried out. Thereafter, a first case-study using an un-tuned Pliocene-Pleistocene benthic oxygen isotope ($\delta^{18}O$) record from ODP Site 846 (Mix et al., 1995; Shackleton et al., 1995) is tested. A second case-study deals with a Danian magnetic susceptibility record from the pelagic carbonate Contessa Highway section in Gubbio, Italy (Sinnesael et al., 2016).

## 2 Methods

### 2.1 Extraction of orbital waveforms

Let's consider a geologic succession that is sampled uniformly in the spatial domain at depths $z_1$, $z_2$ … with $z_0$ being the sampling interval. In an $N$-point measurement analysis frame the stratigraphic record $y(z_n)$ can be represented analytically as a sum of $K$ sinusoids - principal astronomical forcing components $s_k(z_n)$ embedded in the stratigraphic data - plus stochastic perturbation $e(z_n)$:

$$y(z_n) = \sum_{k=1}^{K} s_k(z_n) + e(z_n) \qquad (1)$$

The components responsible for astronomically forced insolation are Earth's orbital eccentricity, obliquity and precession whilst the perturbation accounts for climatic and stratigraphic noise (e.g. Meyers et al., 2008). The sinusoids $s_k(z_n)$ are typically non-stationary, exhibiting some variations in the instantaneous amplitude and phase within the analysis frame:

$y(z_n) = \sum_{k=1}^{K} A_k(z_n) cos[2\pi f_k z_n + \phi_k(z_n)] + e(z_n)$            (2)

where $f_k$ are the average spatial frequencies in cycle m$^{-1}$. The instantaneous amplitude (envelope) $A_k(z_n)$ originates in the solar system dynamics. The instantaneous phase has two components: i) the linear phase term $2\pi f_k z_n$ determined by the corresponding orbital cycle, and ii) phase fluctuation term $\phi_k(z_n)$, chiefly due to climate-sensitive sedimentation. The goal

is to estimate $s_k(z_n)$, $k = 1…K$ from the noisy record $y(z_n)$.

     Strictly speaking, the model as described in (2) cannot be solved because more parameters are present (unknowns, i.e. $A_k(z_n)$ and $\phi_k(z_n)$) than there are measurement points: assuming the frequencies $f_k$ a priori known, there are $2KN$ parameters for $N$ measurements. Although this constraint is too restrictive for the purpose of the present study, it turns out that it can be relaxed by using a concept formalized in the following assumption:

**Assumption 1:** In absence of unconformities, $A_k(z)$ and $\phi_k(z)$ are differentiable functions of $z$ throughout the record.

Differentiability implies continuity and this holds true for records generated by the time-continuous forcing and deposition patterns in undisturbed areas. The principal advantage of this assumption is that (2) can be reformulated such to make the

signal model identifiable. By using the identity $cos(\gamma + \delta) = cos \gamma cos \delta - sin \gamma sin \delta$, (2) can be rewritten as follows:

$y(z_n) = \sum_{k=1}^{K} a_k(z_n) sin(2\pi f_k z_n) + b_k(z_n) cos(2\pi f_k z_n) + e(z_n)$            (3)

with

$a_k(z_n) = -A_k(z_n) sin\big(\phi_k(z_n)\big)$            (4)

$b_k(z_n) = A_k(z_n) cos\big(\phi_k(z_n)\big)$            (5)

Although (2) and (3) – (5) are formally identical, they describe the dynamics of the orbital components in a very different way. The model (2) is highly nonlinear in the sense that the cosine is non-stationary in both amplitude and frequency. The model (3) – (5), however, is non-stationary only in amplitude (i.e. $a_k(z_n)$ and $b_k(z_n)$), with linear argument in the sine and

cosine terms. According to Assumption 1 $a_k(z)$ and $b_k(z)$ are continuous functions in $z$. Accordingly, they can be approximated by spatial polynomials of order $P$ evaluated at $z_n$:

$$\alpha_k(z_n) = a_{k,0} + a_{k,1}z_n + \cdots + a_{k,P}z_n^P \approx a_k(z_n) \qquad\qquad (6)$$

$$\beta_k(z_n) = b_{k,0} + b_{k,1}z_n + \cdots + b_{k,P}z_n^P \approx b_k(z_n) \qquad\qquad (7)$$

The "approximately equal" symbol in the last expressions accounts for possible errors in the polynomial approximation – modeling errors. By inserting (6) – (7) into (3) the following equation is obtained:

$$y(z_n) = \sum_{k=1}^{K}\left(a_{k,0} + \cdots + a_{k,P}z_n^P\right)sin(2\pi f_k z_n) + \left(b_{k,0} + \cdots + b_{k,P}z_n^P\right)cos(2\pi f_k z_n) + \varepsilon(z_n) \qquad\qquad (8)$$

According to the last expression, orbital components can be modeled as a stationary sine-cosine $f_k$ – basis modulated by polynomials. The benefits of such a model for analyzing stratigraphic data are:

1) Both instantaneous amplitude and frequency are simultaneously and compactly characterized by a small number of polynomial coefficients.

2) The number of parameters to estimate is $2K(P+1)$, which can be kept smaller than $N$ by either increasing the number of
20 measurements or reducing the number of components to be estimated.

3) The model is linear-in-parameters and can easily be estimated by means of the linear least-squares procedure, outlined in the Appendix to this paper.

25 Once the model parameters (polynomial coefficients in (8)) are estimated, the orbital component waveforms can be extracted from the stratigraphic data in the following way:

$$\hat{s}(z_n) = \sum_{k=1}^{K}\left(\hat{a}_{k,0} + \cdots + \hat{a}_{k,P}z_n^P\right)sin(2\pi f_k z_n) + \left(\hat{b}_{k,0} + \cdots + \hat{b}_{k,P}z_n^P\right)cos(2\pi f_k z_n) \qquad\qquad (9)$$

30 where the symbol '^' stands for estimate. The principal blocks of the present algorithm are shown in Figure 1.

## 2.2 Sedimentation rate estimation and time axis tuning

Once the estimate of the instantaneous frequency is provided, the sedimentation rate is obtained in a straightforward manner. In what follows, a two-step algorithm is described based on the concept of amplitude-phase decomposition in non-stationary signals (Picinbono, 1997).

5 **2.2.1 Step 1 – Instantaneous phase estimation**

By combining (3) – (5) we obtain:

$$A_k(z_n) = \sqrt{a_k^2(z_n) + b_k^2(z_n)} \tag{10}$$

10 $$\phi_k(z_n) = -atan\left(\frac{a_k(z_n)}{b_k(z_n)}\right), \ k = 1 \cdots K \tag{11}$$

The former $(A_k(z))$ is the true instantaneous amplitude whilst the latter $\phi_k(z_n)$ is the true instantaneous phase fluctuation of the orbital components. The estimated instantaneous phase is readily obtained by means of the previously estimated signal model:

$$\hat{\phi}_k(z_n) = -atan\left(\frac{\hat{\alpha}_k(z_n)}{\hat{\beta}_k(z_n)}\right), \ k = 1 \cdots K \tag{12}$$

**2.2.2 Step 2 – Instantaneous frequency estimation**

By definition, the instantaneous frequency is the time derivative of the instantaneous phase (Cohen, 1995):

$$\hat{f}_k(z_n) = f_k + \frac{1}{2\pi}\frac{d\hat{\phi}_k(z)}{dz}\Big|_{z=z_n} = f_k + \frac{\frac{d\hat{\alpha}_k(z)}{dz}\hat{\beta}_k(z) - \frac{d\hat{\beta}_k(z)}{dz}\hat{\alpha}_k(z)}{\hat{A}_k^2(z)}\Big|_{z=z_n}, \ k = 1 \cdots K \tag{13}$$

Assuming that the sinusoids in the stratigraphic signal model are correctly associated to the astronomical forcing components, the sedimentation rate $r_k(z_n)$ can be estimated for each component:

$$r_k(z_n) = \frac{F_k(kyr^{-1})}{\hat{f}_k(cm^{-1})}, \ k = 1 \cdots K \tag{14}$$

with $F_k(kyr^{-1})$ being the known nominal astronomical forcing temporal frequencies. The mean sedimentation rate is obtained by averaging $r_k(z_n)$ over $k$:

30

$$r(z_n) = \frac{1}{K} \sum_{k=1}^{K} r_k(z_n) \qquad\qquad (15)$$

Recalling the definition of sedimentation rate:

$\quad r(z) = \frac{dz}{dt} \qquad\qquad (16)$

the spatial-temporal conversion is carried out by reformulating (16) as a function of space and then integrating it over space. Finally, the integrals are approximated by means of partial sums:

$\quad t_n = \int_0^z \frac{dz}{r(z)} \approx z_0 \sum_{j=1}^{n} \left( r(z_j) \right)^{-1}, \; n = 1 \cdots N \qquad\qquad (17)$

The last expression gives the time points that mitigate the distortion of the spatial axis due to varying sedimentation rate.

### 2.3 Practical considerations

In the present section, the choice of analysis parameters involved with stratigraphic signal model estimation is discussed.

### 2.3.1 Size of the analysis frame - N

This data-dependent issue is dealt with in numerous areas of signal processing and, to the best of our knowledge, there is no analytical solution. Most approaches are heuristic; however, they are usually quite effective as long as there is some a priori
knowledge about the problem at hand. Depending on the data record, a larger or shorter frame size might be appropriate but no frame size is adequate for all data.

Basically, two constraints must be considered when choosing the number of measurement points for analysis. The lower bound on $N$ is settled by the frequency resolution (i.e. $\frac{z_0}{N}$) needed either to resolve closely spaced sinusoidal components (e.g. the different precession components) or to capture at least one period of the slowest signal component e.g. long-term
eccentricity (i.e. Rayleigh frequency). This piece of information is usually available beforehand.

The upper bound on $N$ depends on the speed of fluctuations of the instantaneous amplitude and frequency along the record. Excepting for some special cases, this is something that we do not know a priori. Accordingly, a reasonable decision is to restrict the choice of $N$ to the lower bound.

### 2.3.2 Selection of the components frequencies - $f_k$

According to (2) $f_k$ are defined as the mean sinusoidal frequencies in the analysis frame i.e. the instantaneous frequency fluctuates around $f_k$. In other words, an orbital component behaves as a narrow-band signal whose bandwidth is much smaller than the Nyquist frequency and most of the spectral energy is clustered around $f_k$. Assuming that the range of the overall frequency variation is known for a given orbital component – which is user defined in the present approach –the mean frequency is associated with the strongest peak in the FFT of the signal in the frame. Although such a simple FFT-based peak-picking algorithm approach might introduce certain bias in the $f_k$ estimate, it is readily compensated by the flexibility of the proposed signal model (Fernando et al., 2004).

### 2.3.3 Degree of polynomials - $P$

This parameter is responsible for capturing variations in the terms $A_k(z_n)sin(\phi_k(z_n))$ and $A_k(z_n)cos(\phi_k(z_n))$ in (4) and (5) respectively. For certain proxy records (up to approximately 50 Ma back in time) variations in the instantaneous amplitude might be inferred from the theoretical model of an orbital component (e.g. numerical models of Laskar et al., 2004; 2011a). However, the instantaneous phase fluctuations are in general unknown beforehand – at best the total bandwidth of an orbital component might be known (Laurin et al., 2016). Accordingly, there is not an analytical way to determine $P$ and once again it is necessary to resort to heuristics.

What is known is that the relationship $N – P$ is not arbitrary; broadly speaking, there is a direct dependence in the sense that larger $N$ requires larger $P$ and vice versa (McAulay and Quatieri, 1986). The reason is that smaller $N$ imposes a quasi-stationarity constraint on the signal – an orbital component behaves almost as a stationary sinusoid with only slightly changing instantaneous amplitude and frequency. In accordance, first order polynomial approximation $P = 1$ is usually enough to properly address the modeling requirements in (9). Larger $N$ implies possibly stronger variations of the underlying sinusoids – the quasi-stationarity assumption has to be dropped and thus larger $P$ is needed. Bearing in mind the aforementioned discussion on the size of analysis frame, $P \in [1 – 3]$ is a reasonable choice for most proxy signals.

### 2.4 Uncertainty Analysis

Providing reliable uncertainty bounds in this estimation approach is challenging. Our model is obtained by a weighted least squares procedure, resulting at the same time in the estimated parameter values and an estimate of the covariance matrix. However, the covariance matrix is only valid if i) there are no model errors, ii) there is no exogenous noise in the system that is also passing through a nonlinear operation. Both elements are shortly discussed below.

Model errors: It is hard to avoid model errors on geological data. These model errors come from the polynomial approximations (6) – (7). Accordingly, larger uncertainties (in this study expressed as standard deviations) are due to a low signal-to-noise ratio and/or fast instantaneous amplitude/frequency changes in the analysis frame. It turns out that model

errors will increase the uncertainty on the estimates. There are two reasons for that: the model errors will increase the level of the residuals, and the residuals are no longer independent of the input (actually, they are uncorrelated).

Noise passing through the nonlinearity: If the disturbing noise is passing through the nonlinearity, the observed noise disturbances at the output of the system are no longer independent of the input. In that case all the classical methods to generate uncertainty bounds fail because these assume explicitly that the disturbing noise is independent of the input. In that case alternatives need to be developed to provide reliable uncertainty bounds, but to the best knowledge of the authors, such methods are not available yet. For that reason, the covariance matrix obtained from the estimation can be used as a first indication, but it should be used with care.

To provide a measure of uncertainty in this study we use Monte Carlo simulations (using the covariance matrix) on the instantaneous frequency and sedimentation rate estimates. This was done through following steps. Primary, it is assumed that the model parameters (polynomial coefficients) are normally distributed correlated random variables characterized by a 2K(P+1) multivariate normal distribution.

1)  Together with the model parameter estimates $\hat{\theta}$ in an analysis frame (A.5), the MATLAB ® least-squares routine "*lscov*" also returns the 2K(P+1) $\times$ 2K(P+1) symmetric positive semi-definite covariance matrix S. This matrix contains the parameter variances on its diagonal; the rest of the elements are parameter covariance's which account on the correlation between the parameters.

2)  By means of the MATLAB ® routine "*normrnd*" a set of random parameters entries were generated by sampling the multivariate normal distribution characterized by $\hat{\theta}$ and *S*.

3)  With these entries the procedure (10) – (15) was carried out and the instantaneous frequency and sedimentation rate estimates in the analysis frame were obtained.

4)  Steps 1) – 3) were run for all the analysis frames along the data record and the estimates were recombined by means of the overlap-add strategy.

5)  Steps 1) – 4) were run 100 times and the depth-variant standard deviations on the instantaneous frequency and sedimentation rate were obtained. The figures in this manuscript and the provided MATLAB ® routines in the supplementary materials report by default one standard deviation (1σ).

The outcome of this process provides a hint on the size of the estimate uncertainties and equally on the spatial distribution of the uncertainty. As explained above, attention has to be payed to the absolute size of the uncertainties. Intervals in the analyzed record where the estimations are less reliable will be visible as peaks in the uncertainties, which is valuable information for interpreting and comparing the results of the analysis as a whole. Also the loss in power of an orbital period compared to the noise level causes peaks in uncertainty. This is because the ACE v.1 model always estimates a signal in every analysis frame, even though the power of a signal does not stand out of the noise level.

## 3 Results and Discussion

### 3.1 Synthetic insolation signal

To illustrate the proposed ACE v.1 model, a case-study is presented on a modified insolation signal. As we are dealing with a synthetic case-study aiming to introduce the mechanisms of the ACE v.1 model, we do not yet implement the uncertainty analysis. The basis of the analyzed signal is the classical 65° N summer (21 June) insolation curve (Milankovitch, 1941) for the last six million years, as modeled by Laskar et al. (2004) (Figure 2A). The periodogram shows the different obliquity and precession related frequencies, which are the components determining the changes in insolation (Figure 2B). Subsequently white and red noise are added to the original signal and made the conversion from the time domain towards the distance domain mimicking a changing sedimentation rate (SR) (Figure 1C). The white noise is normally distributed (Gaussian) with the standard deviation providing the signal-to-noise ratio of 0.09. The red noise is generated as a first-order autoregressive process with the correlation factor equal to 0.999 giving rise to the signal-to-noise ratio of 0.003. To mimic a changing sedimentation rate, the time-series is converted into distance-series. The original time-series has one sample per one thousand years. Initially, a constant SR of 2 cm kyr$^{-1}$ is assumed to make a first conversion to the depth domain. The artificial changing SR is constructed by using the second degree polynomial shown in Fig. 2H-I. In practice, the input signal is interpolated, which increases the sampling rate much above the Nyquist frequency; next, the signal is resampled over the grid established by the SR.

The periodogram of the detrended signal is given in Figure 2D, while its spectrogram is depicted in Figure 2E. The level of white noise in combination with the changing SR has made the identification of the main ~41 kyr obliquity (~1.2 cycle m$^{-1}$) peak less obvious. The precession-related frequencies (~2-2.8 cycle m$^{-1}$) are still clearly present; however the merging of the ~22 and ~24 kyr periods into a single ~23 kyr precession component and the resulting amplitude modulation must be noticed (Figure 2E). The high-power peak around the 0.2 cycle m$^{-1}$ frequency is the result of the introduced red noise.

With the user's a priori knowledge, certain frequencies ranges – bandwidths – in the distance domain are interpreted as to correspond with the astronomical frequencies of obliquity and precession: ~41 kyr obliquity [0.62 – 1.38] cycle m$^{-1}$, ~22 and 24 kyr precession [1.49 – 2.35] cycle m$^{-1}$ and ~19 kyr precession [1.75 – 2.75] cycle m$^{-1}$. It must be pointed out that the model does not deal with the identification of the (orbital) components but that the components waveforms are simulated. The orbital component identification remains with the operator. In this particular case, the precession bandwidths overlap as a consequence of the distortion introduced by the artificial SR. As the characterization of this specific distortion is known, this overlap can be dealt with by letting the bandwidths evolve along the record in function of the defined polynomial related to the SR change. However, in case a similar argumentation cannot be ensured for a case-study, the use of the model in its current form is not recommended.

Providing these frequency ranges for this case-study and a frame size of 5 m the model estimates the three given components (Figure 2F). As described in previous chapter, the choice of the frame size is a heuristic trade-off between: (1)

have at least some periods of your lowest frequency component in one frame and (2) be long enough to be able to separate closely spaced components. In this case, the obliquity period of 41 kyr [~1.15 cycle m$^{-1}$] is considered as the lowest frequency to be taken into account. The aim is not to separate the ~22 and ~24 kyr precession components, but distinguish the merged ~23 and the single ~19 kyr periods.

5      From the modeled waveforms, the estimated instantaneous frequencies are extracted (Figure 2G). To avoid potential discontinuities at the frame edges, the well-known overlap-add method is used (e.g. Verhelst, 2000). Essentially, overlapping data are summed and afterwards normalized again by a given weight, which is a function of the number of frame overlaps for a data point.

     Using the a priori knowledge, the frequencies can be converted into SR estimates (Figure 2H). Taking the average SR of 10   all three modeled components (Figure 2I), the distance series can be transposed into a time-series. The periodogram of the new time-series still contains traces of the white and red noise but is much cleaner than the modified insolation signal (Figure 2J).

     In summary, it is the user who needs to identify and provide a bandwidth for each component that is to be modeled and the size of the analysis frame. The selected bandwidths are typically based on other available geological constraints, like bio- or 15   magnetostratigraphy. This artificial case-study illustrates the good performance of the model in estimating components, extracting the instantaneous frequencies and making a successful conversion towards the time domain. This conversion has been done autonomously by the algorithm for given frequencies that have to be traced. No band-pass-filtering or tuning to another form of model has been used. Here, the test started from a time-series of insolation with very well constrained orbital components. The following two case-studies deal with real geological case-studies.

## 3.2 ODP 846 benthic δ$^{18}$O record

The stratigraphy and chronology of the Plio-Pleistocene has benefited largely from benthic oxygen isotope records and its stacks (Lisiecki and Raymo, 2005; Huybers, 2007; Hilgen et al., 2012 *in* Gradstein et al., 2012). One of the longest and most detailed records of this sort comes from Ocean Drilling Program (ODP) Site 846 from the tropical Pacific Ocean (3°06'S, 25   90°49'W; Mix et al., 1995; Shackleton et al., 1995). The length of the core is ~206 m, spanning the last 5.3 Ma and is sampled for benthic δ$^{18}$O with an average resolution of 10 cm (~2.5 kyr; Mix et al., 1995; Shackleton et al., 1995). Spectral analysis has identified strong ~100 kyr and ~41 kyr periodicities. While the origin of the 100 kyr cycle in the Pleistocene glacial is subject of debate (Imbrie et al., 1993; Muller and MacDonald, 1997; Lisiecki, 2010), the 41 kyr periodicity can be attributed to obliquity. Other studies on ODP 846 alkenone-derived sea surface temperatures have shed light on the relative 30   importance of obliquity and precession (high and low latitude) on the climatology of the tropical ocean system (Liu et al., 2004; Cleaveland and Herbert, 2007; Herbert et al., 2010). The ODP 846 benthic δ$^{18}$O record is used in the reference LR04 Pliocene-Pleistocene benthic stack (Lisiecki and Raymo, 2005).

This record is selected for a case-study as it is well studied and has an astronomically calibrated age model (LR04), which allows for the evaluation of the proposed approach. The raw data is the benthic $\delta^{18}O$ depth series (Figure 3A). After detrending, the peridiogram and spectrogram reveal elevated spectral power around ~0.3 and ~0.6 cycle $m^{-1}$ (Figure 3B, 3C). Using the a priori knowledge of the core, the first dominant frequency is identified as corresponding with a 100 kyr

periodicity, while the ~0.6 cycle $m^{-1}$ corresponds with the 41 kyr obliquity. Figure 3D shows the spectrogram of the two modeled components. Following a similar reasoning as in the previous case-study, an analysis frame size of 10 m is utilized. Using the instantaneous frequency estimates (Figure 3E), the corresponding evolution of the sedimentation rate can be estimated (Figure 3F). As the origin of the 100 kyr cycle is debatable (Imbrie et al., 1993; Muller and MacDonald, 1997; Lisiecki, 2010), the 41 kyr derived sedimentation rate is selected and compared with the SR as derived from the LR04 age

model for the ODP 846 core (Figure 3I). To do so, the stratigraphic levels are subtracted of subsequent samples and divided by their age (LR04) difference (Figure3I). This simple operation results in abrupt changes in sedimentation rate which cannot be captured by the ACE v.1 model approach. These abrupt changes originate from the principle that the LR04 age model is based on a stack, which enhances the signal-to-noise ratio compared to an analysis on a single core. Moreover, it is tuned to an ice model that is driven by the 21 June insolation at 65°N. Therefore, in this study a 500-point running average is

taken to smooth the results (Figure 3I). Only the long term averaged trends are captured by our ACE v.1 estimation approach and not the fine-scaled sedimentation rates of the original LR04 age model. This is because: (i) the ACE v.1 model in its current form cannot deal with fast changes in sedimentation rate (ii) the signal-to-noise-ratio of the stacked LR04 age model is superior to the ratio of a single record and (iii) no other a priori information as that the obliquity band should be in a certain frequency range is used in the ACE v.1 analysis, contrary to the LR04 age model where other geological constraints

are included. In the described approach, a frequency range is selected, which with the user's a priori knowledge could be associated with the 41 kyr obliquity [0.47 – 0.71] cycle $m^{-1}$ and let the algorithms extract its waveform. The corresponding age model is then created by using the estimated instantaneous frequency changes and the association with the 41-kyr obliquity period.

Except for a small difference between 60-90 m the match between the results of the ACE v.1 model and the averaged

LR04 age model is close. Remarkably, the interval has a pronounced elevated uncertainty on its estimation (Figures 3G and 3H). The origin of the mismatch is the lower signal amplitude in this interval. Note the elevated signal amplitude around 70-80 m around the frequency of 0.5 cycle $m^{-1}$ whereas the 41-kyr component is mainly around the 0.6 cycle $m^{-1}$ (Figure 3C). This feature makes that the ACE v.1 model suggests a small drop in sedimentation rate between 70-80 m (Fig. 3I), whereas the LR04 age model suggests a gradual rise in sedimentation rate in this interval. This mismatch could be reduced by

increasing the lower boundary of the selected frequency range for the obliquity component estimation. A similar feature on a smaller scale is detectable around level 40-50 m (Figure 3G). Between 140 m and the bottom of the core, there is low power in the 41-kyr frequency range. However, in contrast to the 60-90m interval, there are no neighboring (in the frequency domain) elevated signal amplitudes. The loss of power in this frequency range at this position translates in an elevation of the uncertainty on the estimations for this interval (Figures 3G and 3H). In general, the comparison between the ACE v.1

modeled sedimentation rates and LR04 sedimentation rates yield satisfactory results, as stratigraphic intervals where a mismatch exists are red-flagged by increased uncertainty on the ACE v.1 model. The sign of the mismatch in sedimentation rates is not consistent throughout the core, which means that over- and underestimates of sedimentation rates cancel each other out. In this particular case study, there seems to be an overestimation in total duration (compared to the LR04 age model) of ~5 %, which is significant but not so bad, considering that we simply used the tracking of the obliquity related frequency. The periodogram of the tuned record, based on the 41 kyr derived SR, reveals a much cleaner signal (Figure 3J). Using the comparison between the SR derived in this novel approach and deduced from the LR04 age model for the ODP 846 record (Figure 3I), the conclusion is that again the algorithm performs well in capturing the main averaged trend.

## 3.3 Danian magnetic susceptibility record

This second case-study is related to a Danian magnetic susceptibility (MS) record from the pelagic carbonate Contessa Highway section, Gubbio, Italy. The Gubbio sections are well-known for its pioneering studies including planktonic biostratigraphy (Luterbacher and Premoli Silva, 1964), magnetostratigraphy (Arthur and Fischer, 1977) and the Cretaceous-Palaeogene boundary (Alvarez et al., 1980). The MS record consists of a total of 1049 samples with a sample spacing of 1 cm for the lower half of the record and a sample spacing of 2 cm for the upper half (Sinnesael et al., 2016). The total length is 14 m and includes a large portion of the Danian including magnetochrons C29r (Paleogene part) to the top of C27n (Lowrie at al., 1982, Coccioni et al., 2012a, b). Sinnesael et al. (2016) presented a cyclostratigrahic framework for this section, reporting the orbital imprint in this dataset.

Before the actual analysis, anomalous peaks in MS which are typically related to volcanic ashes (Figure 4A) are removed. Also, the first 1.3 m of the record was excluded from analysis because the MS record is heavily disrupted by the Dan-C2 hyperthermal event (Coccioni et al., 2010). The periodogram of the detrended signal shows elevated power at frequencies of 2.3 and 6.0 cycle m$^{-1}$ (Figure 4B). With a sedimentation rate estimate derived from bio- and magnetostratigraphic constraints from the Danian in Gubbio of 4 m Myr$^{-1}$ (Coccioni et al., 2012b), these frequencies respectively correspond with the periods of short eccentricity and obliquity. Because of tidal dissipation, the periodicity of obliquity cycles during the Danian was shorter than today's ~41 kyr value. Here, a duration of 39.6 kyr per obliquity cycle is used as calculated by Berger et al. (1992). Accordingly, three frequency ranges are traced, which respectively seem to correspond with an orbital period: short eccentricity [1.78 – 2.57] cycle m$^{-1}$, obliquity [5.13 – 6.58] cycle m$^{-1}$ and precession [8.98 – 10.98] cycle m$^{-1}$ (Figure 4D, 4E). Except for two intervals (102-104 m and 108-110 m), the precessional component does not exceed the noise level (Figure 4C). Nevertheless, SR is derived using all three astronomical components, all showing a similar pattern of a more or less stable SR around 4.4 m Myr$^{-1}$ in the lower part of the section. From 111 m till the end of the section at 114 m, a transition towards higher SR (4.9 m Myr$^{-1}$) occurs. Notice an increase in uncertainty at this transition (Figures 4G and H). The final uncertainties on the SR in this case-study (one standard deviation) stay however very small, several orders of magnitudes smaller than in the ODP846 case-study. The sedimentation rate estimate that is based on precession shows a

somewhat deviating pattern between 104-108 m, corresponding to the interval where this component only has a minor imprint (Figure 4C and E). The average SR is used to transform the distance-series into a time-series. The periodogram of the tuned time-series shows a clear obliquity peak and elevated power in the range of the 100 kyr eccentricity (Figure IG). Interestingly, the largest peak of the new spectrum is close to 405-kyr eccentricity period. Moreover, the lower frequency domain of the spectrum contains fewer peaks that are unrelated to astronomical forcing. Frequencies in the domain of the precession stay hardly distinguishable from the noise levels.

In the original publication, Sinnesael et al. (2016) used a band-pass filter to extract the long eccentricity component, assuming a constant average sedimentation rate of 4.3 m Myr$^{-1}$. Subsequently, this filtered signal was used to tune the record to the eccentricity solution (Laskar et al., 2011a). The results reported in this study are thus in agreement with the original interpretation for the part of the section before the jump in SR towards a value of ~ 4.9 m Myr$^{-1}$ (Figure 4J). However, this slight increase in SR was not reported by Sinnesael et al. (2016). Interestingly, a similar increase in sedimentation rate has been described in a cyclostratigraphical study of a nearby coeval section in the Bottaccione Gorge, also in Gubbio, Italy (Galeotti et al., 2015). Galeotti et al. (2015) attribute this change to the recovery of the carbonate productivity, which had dropped after the Cretaceous-Boundary boundary (K-PgB) environmental changes. Differently, Galeotti et al. (2015) obtain SR lower than 4 m Myr$^{-1}$ (up to 2 m Myr$^{-1}$) for the interval between 1 and 4 m above the K-PgB (Figure 4J). The first 1.3 m of the Danian was not taken into account in the analysis on the Contessa section, because of the perturbation of the DAN-C2 event on the MS signal. However, for the overlapping stratigraphic levels, using this new modeling approach, there are no indications for such a significant drop in SR. Also, Sinnesael et al. (2016) do not observe a drop in SR in this stratigraphic interval. The possible presence of slump structures in the stratigraphic interval above 3-4 m above the K-PgB in the Bottaccione section could be an explanation for this different interpretation.

The constraint of the use constant band-pass filter ranges for a (sub-) record disappears with the ACE v.1 modelling approach. This reduces the risk of missing potential delicate changes in SR as for example the small increase towards the end of this particular section.

## 4 Conclusions and Prospects

This study introduces a new approach to time-series analysis in the field of cyclostratigraphy. The main focus is on the estimation of already identified components in a signal. The identification (or detection) of the components is based on the a priori knowledge of the user, given available geological constraints. Those components are given a frequency-range, which is determined by the user and can correspond with astronomical periods of for example eccentricity, precession and obliquity. Once this first step has been taken, the selected components are simulated by making use of a model that is not based on FFT-derived methods, bandpass filtering or other commonly used methods in cyclostratigraphy but one that relies

on polynomial modeling. Basic uncertainty analysis is provided too in order to be able to assess the size and distribution of the model estimate uncertainties.

The ACE v.1 model in its current version should be used under following constraints: (1) significant sinusoidal-like variations can be detected and identified in a proxy record, (2) the frequency ranges of the components cannot overlap for a given analysis, (3) the user makes an appropriate tradeoff between the analysis frame size and the degree of the used polynomials and (4) no fast changes in sedimentation rate can be detected.

The first case-study on a controlled modified insolation signal documents the ability of this proposed new approach to successfully model components of a signal. Furthermore, it shows that the estimated instantaneous frequencies derived from the modeling can easily be converted to sedimentation rates which can be used to convert distance-series into time-series. A more robust result is obtained by taking the average sedimentation rate of the different estimated components. New spectral analysis shows strongly reduced levels of both white and red noise.

Also the second case-study concerning the benthic $\delta^{18}O$ record of ODP Site 846 illustrated key features of the proposed ACE v.1 model. The tracking of the obliquity component is successful and the derived sedimentation rates on the basis of this modeled component are in close agreement with the average age model for ODP846 as constructed by LR04. However, only the long term averaged trends are captured and not the fine-scaled sedimentation rates of the original LR04 age model. This is because: (a) the ACE v.1 model in its current form cannot deal with fast changes in sedimentation rate (b) the signal-to-noise-ratio of the LR04 is superior to the ratio of a single record and (c) no other a priori information coming from other geological constraints is included as that the obliquity band should be in a certain frequency range. Moreover, the practical use of the uncertainty analysis on the size and distribution of the model estimates is well exemplified. This case-study illustrates too that the suggested approach is an additional instrument in time-series analysis to automatically extract waveforms and derive time. As such one can exclude the human influence in the tuning process (often heuristically determining relative minima and maxima in a proxy record). However, the verification with available geological constraints remains essential in the validation of the model.

The third case-study dealt with a Danian magnetic susceptibility record from the Contessa Highway section Gubbio, Italy. It forms a good illustration of how a priori knowledge is used to select certain components to model, but also how the results of the model can be coupled back to existing interpretations. Moreover, it demonstrates the disadvantages of using the classical bandpass filtering approach in the tuning process, which is circumvented with this new modeling approach. The approach sheds new light on the astrochronological age model and derived sedimentation rates of the Danian in Gubbio. Following the model, there is probably no significant drop in SR in the oldest half of the Danian section in Gubbio, but there is a suggested increase in SR from the middle of magnetochron C27r. Again the crucial role of the verification of the model results with other available geological constraints must be emphasized, in this case as discussed in Galeotti et al. (2015) and Sinnesael et al. (2016).

This paper convincingly introduces the principles and illustrates the functioning of the concept of time-series analyses by polynomial modeling of sinusoidal behavior in earth-science proxy series. Further work will concentrate on: (i) automated

identification of significant components, (ii) the release of the constraint of non-overlapping of the component's frequency ranges and (iii) allowing the tracking off faster sedimentation rate changes. Note, that this modeling approach is designed to be used in a cyclostratigraphical framework. However, it can easily be adapted for all kind of time-series analyses on data, which contain significant cyclic (sinusoidal) variation. The reader is invited to make use of the ACE v.1 model and provide

feedback on its functioning and further development.

**Code Availability**

The ACE v.1 model is designed in MATLAB ® and all scripts are available in the Supplementary Materials. These include a main script that enables the user to load data, define model parameters, use the functions and produce basic graphical output. Also the separate scripts of the three functions "OrbitalComponentEstimation", "SedimentationRateEstimation" and "UncertaintyAnalysis"               are                included             as              well              as              an              instructive              manual "Manual_ACEv1_Model_Sinnesael_etal_2016_GMD.txt".

**Acknowledgments**

Matthias Sinnesael thanks the Research Foundation – Flanders (FWO) for the awarded PhD fellowship (FWOTM782). David De Vleeschouwer was funded through European Research Council (ERC) Consolidator Grant "Earthsequencing" (Grant Agreement No. 617462). This work was supported in part by the Fund for Scientific Research (FWO-Vlaanderen

Grant G009113N and support to Johan Schoukens), by the Flemish Government (Methusalem METH1 to Johan Schoukens), and by the Belgian Government through the Inter university Pole of Attraction (IUAP VII) Program (P7/15 PLANET TOPERS and P7/19 DYSCO).

**Appendix**

**Estimation of the model parameters**

The model parameters in (8) are estimated by minimizing the following cost function in the matrix form in the least-squares sense:

$$min(\mathbf{y} - \mathbf{H\theta})^T(\mathbf{y} - \mathbf{H\theta}) \qquad (A.1)$$

where the symbol $T$ denotes matrix transpose operator. The elements in (A.1) are defined as follows:

- Vector containing the stratigraphic data

$$\mathbf{y} = \left(y(z_1), y(z_2), \cdots, y(z_N)\right)^T \qquad (A.2)$$

- Vector containing the model parameters

$$\mathbf{\theta} = \left(a_{1,0}, \cdots a_{1,P}, b_{1,0}, \cdots b_{1,P} \cdots a_{K,0}, \cdots a_{K,P}, b_{K,0}, \cdots b_{K,P}\right)^T \tag{A.3}$$

- Matrix containing the signal model

$$\mathbf{H} = \left(\mathbf{H}_{1,s}\ \mathbf{H}_{1,c}\ \mathbf{H}_{2,s}\ \mathbf{H}_{2,c}\ \cdots \mathbf{H}_{K,s}\ \mathbf{H}_{K,c}\right) \tag{A.4}$$

with

$$\mathbf{H}_{k,s} = \begin{pmatrix} sin(2\pi f_k z_1) & z_1 sin(2\pi f_k z_1) \cdots z_1^P sin(2\pi f_k z_1) \\ sin(2\pi f_k z_2) & z_2 sin(2\pi f_k z_2) \cdots z_2^P sin(2\pi f_k z_2) \\ \vdots & \vdots \qquad\qquad \vdots \\ sin(2\pi f_k z_N) & z_N sin(2\pi f_k z_N) \cdots z_N^P sin(2\pi f_k z_N) \end{pmatrix}$$

$$\mathbf{H}_{k,c} = \begin{pmatrix} cos(2\pi f_k z_1) & z_1 cos(2\pi f_k z_1) \cdots z_1^P cos(2\pi f_k z_1) \\ cos(2\pi f_k z_2) & z_2 cos(2\pi f_k z_2) \cdots z_2^P cos(2\pi f_k z_2) \\ \vdots & \vdots \qquad\qquad \vdots \\ cos(2\pi f_k z_N) & z_N cos(2\pi f_k z_N) \cdots z_N^P cos(2\pi f_k z_N) \end{pmatrix}$$

The solution to the least-squares problem (A.1) is the vector of sought model parameters:

$$\widehat{\mathbf{\theta}} = \mathbf{H}^\dagger \mathbf{y} \tag{A.5}$$

where $\mathbf{H}^\dagger$ is the pseudo inverse of $\mathbf{H}$.

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

**Figures**

**Figure 1:** (*a*) **Major steps in cyclostratigraphic signal processing.** (*b*) **The estimation step – a focus of the present study.**

**Figure 2: ACE v.1 analysis for a synthetic insolation signal. (A) 6 Ma Insolation for 65°N 21 June (Wm$^{-2}$) (Laskar et al., 2004). (B) The periodogram of the clean insolation signal as plotted in Fig. 2A. (C) To make the conversion from the distance towards the time domain a constant sedimentation rate of 2 cm kyr$^{-1}$ is used. Using linear interpolation a non-constant sedimentation rate is**
**mimicked. The changing sedimentation rates is plotted in Figs. 2H-I. Additional white and red noise are added to the signal as discussed in the text. (D) The periodogram of the modified insolation signal as plotted in Fig. 2C, which is noisier than the original periodogram (Fig. 2B) because of the added perturbations. (E) The spectrogram of the modified insolation signal. (F) The spectrogram of the three modeled components (41 kyr obliquity, 23 kyr precession and 19 kyr precession) using ACE v.1. (G) Estimated instantaneous frequencies for the three modeled components. (H) Using the associated astronomical periods of the**
**modeled frequencies, the corresponding sedimentation rate is calculated and compared with the initial input change in sedimentation rate. (I) Comparison between the initial input change in sedimentation rate and the mean of the modeled components. (J) Periodogram of time-series of the signal where the mean sedimentation rate estimate is used to make the conversion to the time domain.**

**Figure 3: ACE v.1 analysis for a Plio-Pleistocene benthic oxygen isotope record. (A) ODP Site 846 δ$^{18}$O record (Wm$^{-2}$) (Mix et al., 1995; Shackleton et al., 1995). (B) The periodogram of the ODP Site 846 δ$^{18}$O record as plotted in Fig. 2A. (C) The spectrogram of the ODP Site 846 δ$^{18}$O record. (D) The spectrogram of the two modeled components (100 kyr periodicity and 41 kyr obliquity) using ACE v.1. (E) Estimated instantaneous frequencies for the two modeled components. (F) Using the associated (astronomical) periods of the modeled frequencies, the corresponding sedimentation rate is calculated. (G) The uncertainties on the estimated**
**instantaneous frequencies (1σ). (H) The uncertainties on the sedimentation rate estimates (1σ). (I) Comparison between the ACE**

v.1 estimated sedimentation rate and the derived non-averaged and 500-point averaged sedimentation rates for ODP Site 846 according to the age model of the LR04 stack (Lisiecki and Raymo, 2005) and the estimated change in sedimentation rate based on the modeled obliquity component. (J) Periodogram of time-series of the signal where the mean sedimentation rate estimate is used to make the conversion to the time domain.

Figure 4: ACE v.1 analysis for a Danian magnetic susceptibility record. (A) Contessa Highway Danian magnetic susceptibility record ($m^3$ $kg^{-1}$) (Sinnesael et al., 2016). (B) The periodogram of the Danian magnetic susceptibility as plotted in Fig. 2A. (C) The spectrogram of the Danian magnetic susceptibility record. (D) The spectrogram of the three modeled components (100 kyr eccentricity, 39.6 kyr obliquity and 22.5 kyr precession using ACE v.1 (Berger et al., 1992). (E) Estimated instantaneous frequencies for the three modeled components. (F) Using the associated (astronomical) periods of the modeled frequencies, the corresponding sedimentation rate is calculated. (G) The uncertainties on the estimated instantaneous frequencies (1σ). (H) The uncertainties on the sedimentation rate estimates (1σ). (I) Periodogram of time-series of the signal where the mean sedimentation rate estimate is used to make the conversion to the time domain. (J) Comparison between the ACE v.1 estimated sedimentation rate and the derived sedimentation rates in the studies by Galeotti et al. (2015) and Sinnesael et al. (2016).

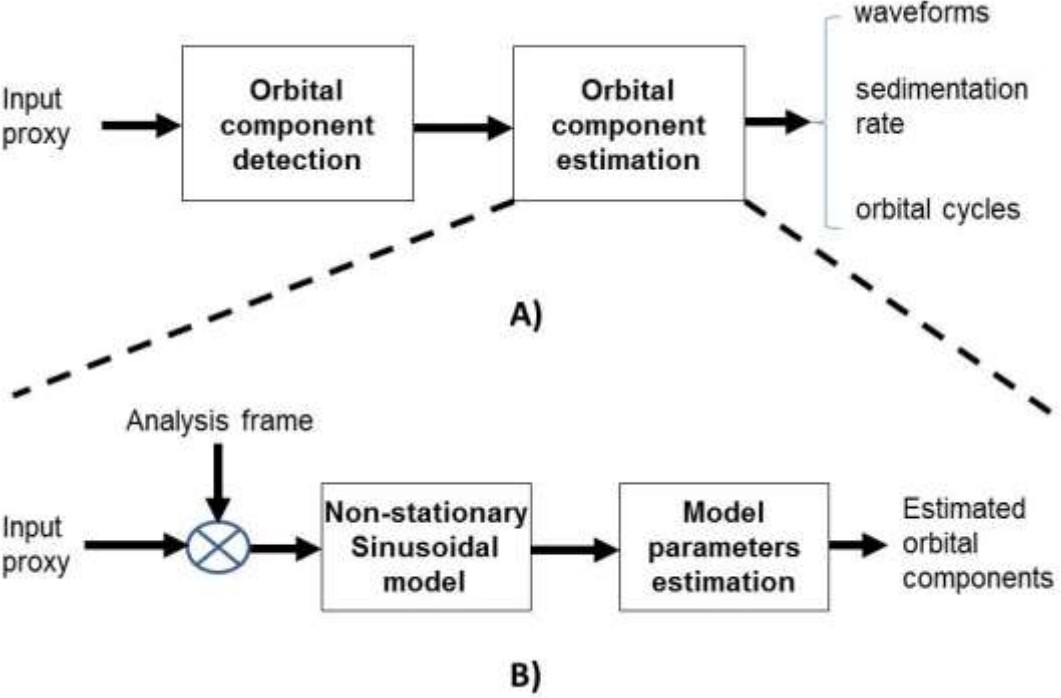

A)

B)

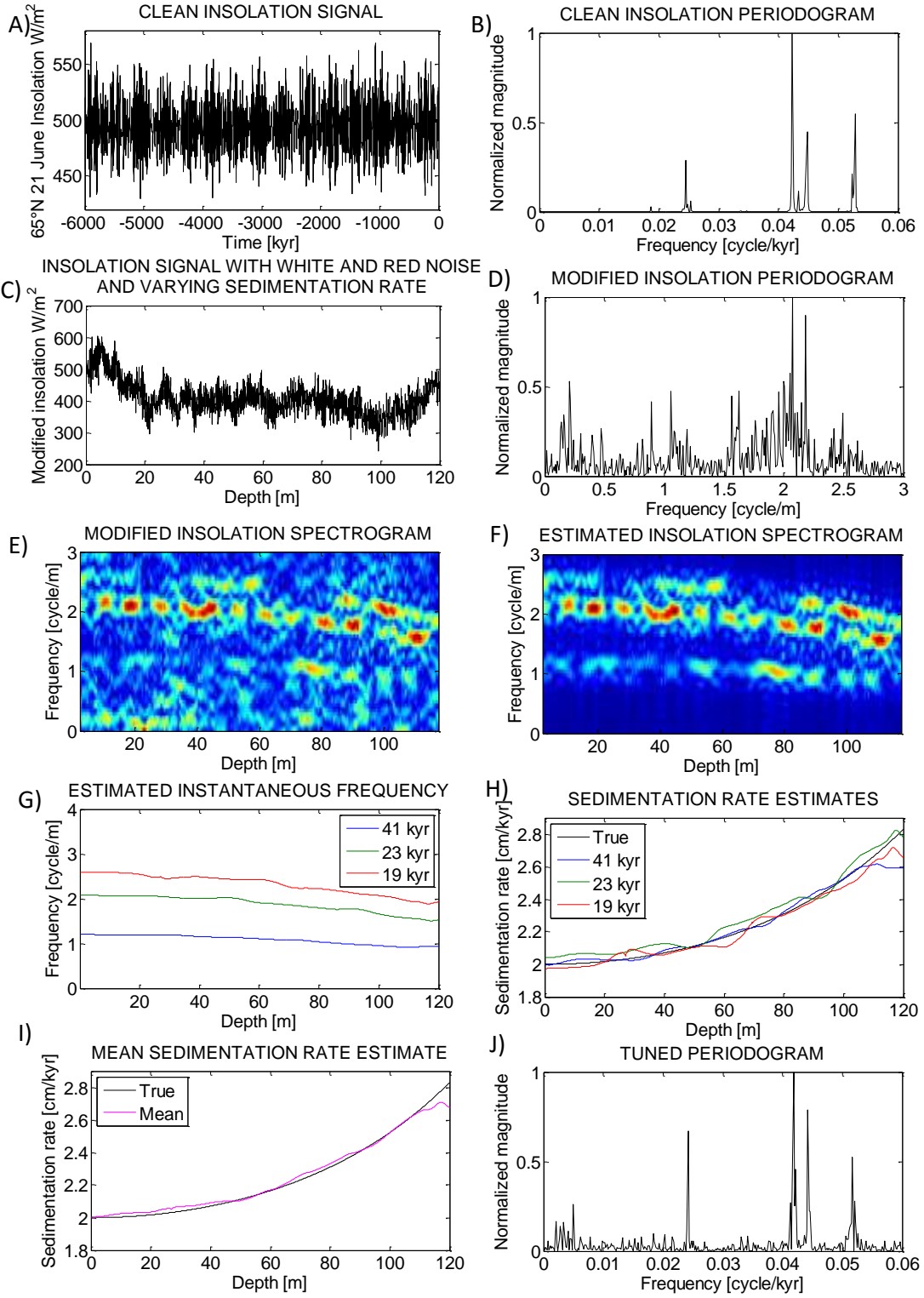

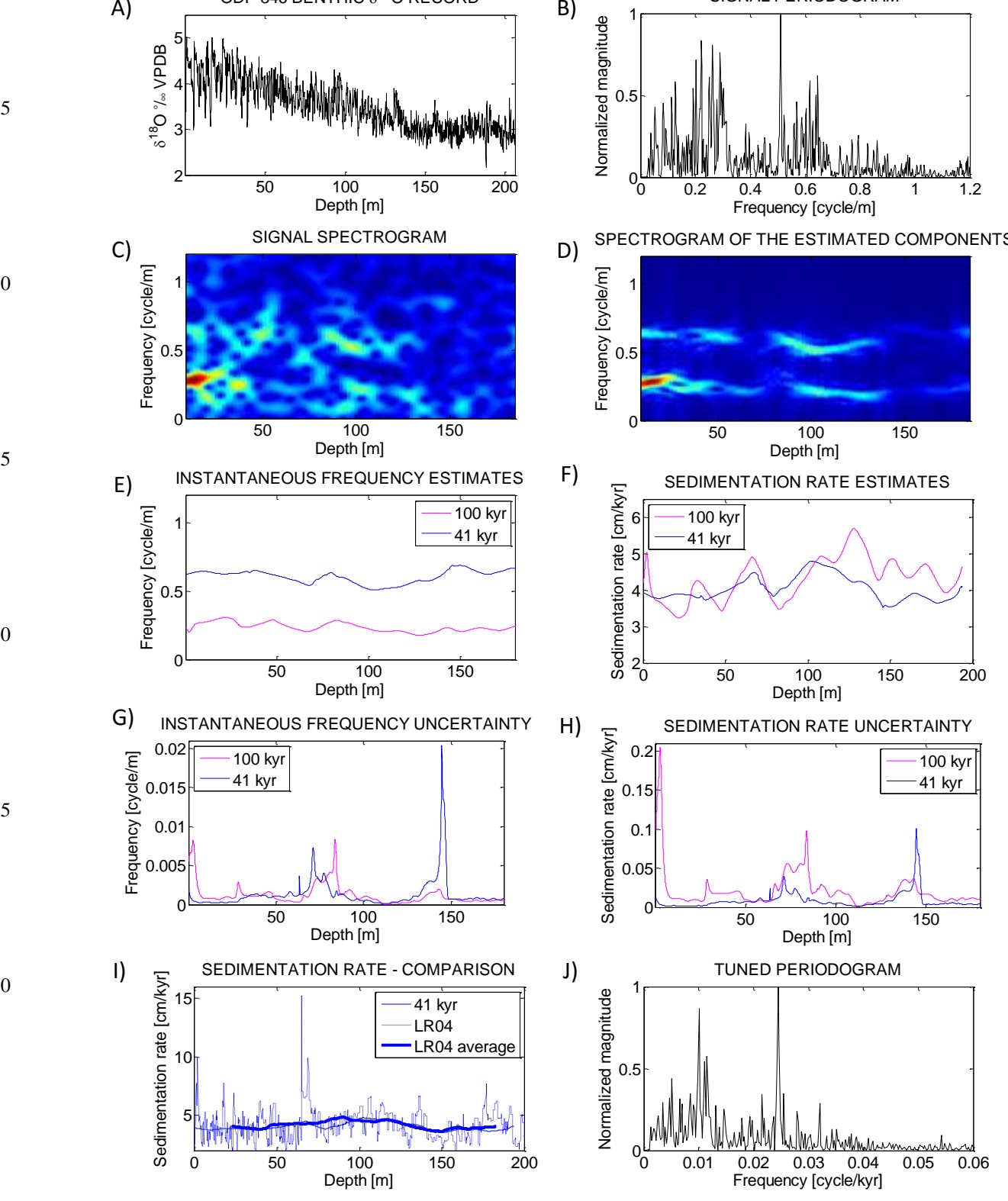

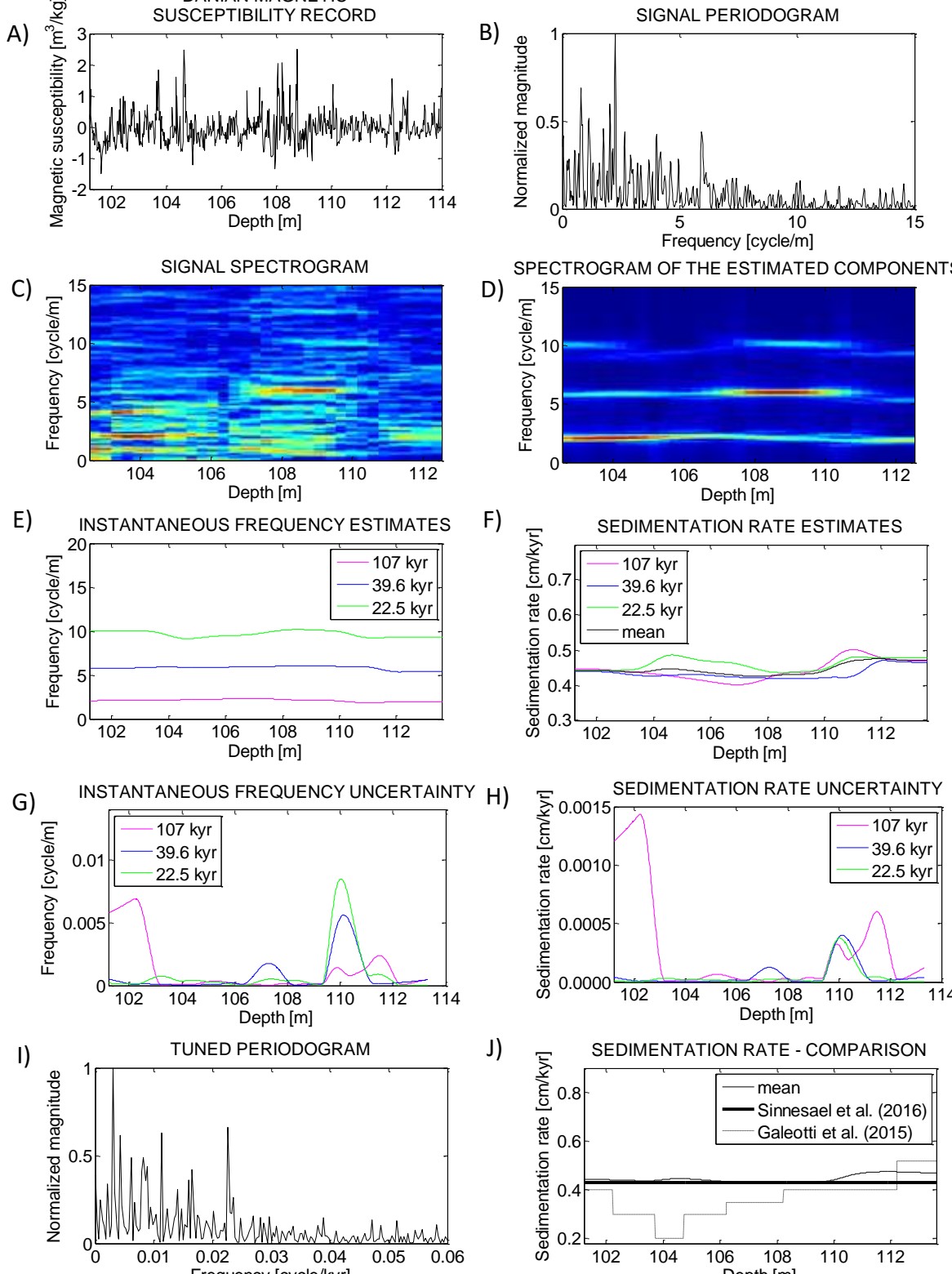