# Peer review of "Astronomical component estimation (ACE v.1) by time-variant sinusoidal modeling."

_Geoscientific Model Development, 2016_

## Referee Comment (RC1) · Anonymous Referee #1 · 1 Jul 2016

\*General comments

The manuscript describes the model ACE v.1 (Astronomical Component Estimation, version 1), consisting on a linear sum of time-varying sinusoidal components with preset frequencies and with amplitudes described by low-order polynomials. The presentation is clear, concise, and scientifically sound. The depicted parametric approach can be relevant not only in cyclostratigraphy but also in the wider context of general time series analysis. Therefore in my opinion the manuscript can be published after some minor revisions.

\*Specific comments

My main concern in the proposed approach is the lack of any measure of uncertainty associated with the estimation procedure. Error bars for the parameters estimated by

linear least squares should be easily obtainable, and included in the derived instantaneous amplitude and frequency results.

Fig. 1 (b) - the role of the "frame" in the figure is not clear

Fig. 3 (G) - the width of the line representing 41 kyr is not consistent

*Technical corrections

Page 5, line 26: principle

Page 10, line 30: maybe rephrase, the sentence is not clear ("Using the estimated instantaneous frequencies age model is derived")

Page 11, line 1: siganl

Page 13, line 29: tracing off

---

## Referee Comment (RC2) · Anonymous Referee #2 · 7 Jul 2016

This manuscript present a method for better constraining age-models in the context of cyclostratigraphy. The paper is clear and short, the methodology is well exposed and scientifically sound. I therefore believe it deserves being published after some minor corrections discussed below.

1/ As also mentionned by the first reviewer, I believe it is critical to provide and discuss the uncertainties associated with the fitting procedure, and try to translate them into error bars in the final age models or sedimentation rates.

2/ I am also a bit frustrated by the lack of discussion on the results obtained with the ODP846 record (basically Fig.3G) which is only presented by the sentence : Âń Except for a small difference between 70 and 100m the match is close Âż. How large is the mismatch in terms of absolute age ? Where does this mismatch actualy come

from ? There is a lower signal amplitude at 41kyr at this time (Fig.3D), but this is also the case below 150m where the agreement with LR04 is rather good . . . The LR04 sedimentation rate is rather flat, so there is (a priori) no strong change in the record at this time. The explanations given in the conclusion (page 13 lines 10-16) are therefore not fully convincing for this particular case study. 3/ Clearly, the Danian record is the best example of the added value of the method (since there are little stratigraphic constraints except cyclostratigraphy). The advantages of the new method are discussed in the text (page 13, lines 17-25) but not well illustrated on the figures. It would be quite easy, and very helpful, to add on Figure 4F indications of alternative sedimentation rates (traditionnal tuning by Sinnesael et al 2016, . . .) somewhat equivalent to Figure 3G.

―――――――――――――――――――――

---

## Author Comment (AC1) · 25 Jul 2016

Sinnesael et al. appreciate the review and suggestions for the discussion paper "Astronomical component estimation (ACEv.1) by time-variant sinusoidal modeling" by two anonymous referees.

In this document we provide a point-by-point answer to the interactive comments. These comments will be taken into account while preparing the final version of this manuscript.

**Anonymous Referee #1**

**\*Specific comments**

**1/ My main concern in the proposed approach is the lack of any measure of uncertainty associated with the estimation procedure. Error bars for the parameters estimated by linear least squares should be easily obtainable, and included in the derived instantaneous amplitude and frequency results.**

Providing a measure for uncertainty on the estimations is the main concern for both referees. We recognize that in the current version of the algorithm- uncertainties related to the sinusoidal fitting procedure are not taken into account. We want to meet this concern by discussing it in the methodology and by implementing a measure for the uncertainty in the ACEv.1 MATLAB routines. We are currently exploring different techniques to estimate that uncertainty and to translate this uncertainty into sedimentation rate (and thus geologic time).

A first possibility to address this would be to use the uncertainties (standard deviations from the least-squares) off the parameters that are used to estimate the waveforms. However, the non-linear relationship between the parameters and instantaneous frequency (sedimentation rate), implies no closed-form solution for the uncertainties. For a single data record, the number of parameters to estimate = (Number of orbital components) x (Polynomial order + 1) x 2 x (Number of analysis frames). Combining all uncertainties of the parameters with error propagation and in our case of overlapping (analysis frames) would be computationally expensive.

Therefore, we will adopt a more time-efficient and equally elegant alternative approach. This approach consists of Monte-Carlo simulations of the jitter on proxy measurements, as well as on the depth/time scale. Such approach allow for the evaluation of the robustness of the sinusoidal fit.

**2/ Fig. 1 (b) - the role of the "frame" in the figure is not clear**

We will redraft Fig. 1b so that the different concepts illustrated are more readily clear to the reader (for example the role of the "frame", which is called the "analysis frame" in the text of the manuscript). Moreover, we will make sure that the words "analysis frame" and "window" are consistently used throughout the whole manuscript, so to improve clarity.

**3/ Fig. 3 (G) - the width of the line representing 41 kyr is not consistent**

We agree it makes more sense to have the 41 kyr line thinner and thicken the line representing the averaged (500 point) sedimentation rates which were derived from the LR04 age model. We will adopt this suggestion.

**\*Technical corrections**

Thank you, these will be corrected.

**Anonymous Referee #2**

**\*Specific comments**

**1/ As also mentionned by the first reviewer, I believe it is critical to provide and discuss the uncertainties associated with the fitting procedure, and try to translate them into error bars in the final age models or sedimentation rates.**

See the written answer for "1/ Anonymous Referee #1", same remark.

**2/ I am also a bit frustrated by the lack of discussion on the results obtained with the ODP846 record (basically Fig.3G) which is only presented by the sentence : " Except for a small difference between 70 and 100m the match is close". How large is the mismatch in terms of absolute age ? Where does this mismatch actually come from ? There is a lower signal amplitude at 41kyr at this time (Fig.3D), but this is also the case below 150m where the agreement with LR04 is rather good ... The LR04 sedimentation rate is rather flat, so there is (a priori) no strong change in the record at this time. The explanations given in the conclusion (page 13 lines 10-16) are therefore not fully convincing for this particular case study.**

Reviewer #2 wishes for more discussion of the technical aspects of the ODP846 case study. We recognize that in the original version of the manuscript, this discussion is rather limited, and we intend to elaborate on this topic in the revised version. Below, we break down the comment into pieces and provide a point-by-point answer.

**I am also a bit frustrated by the lack of discussion on the results obtained with the ODP846 record (basically Fig.3G) which is only presented by the sentence : " Except for a small difference between 70 and 100m the match is close".**

The mismatch is also discussed in the conclusions, however it is indeed better to rearrange the text and provide more in-depth information in the discussion itself and less in the concluding chapter. Hence, a restructuring of this discussion is planned for the revised version of the manuscript. See more in the points below.

**How large is the mismatch in terms of absolute age ?**

This depends on the position down the core. Sometimes the 41 kyr estimate derived sedimentation rates will be higher than the LR04 sedimentation rates, sometimes lower. This also means over-and underestimations cancel each other out throughout. We will discuss the difference in total duration of both approaches, as well as the maximum and minimum discrepancy, in the discussion of the manuscript.

**Where does this mismatch actually come from ?**

This is mentioned in the conclusions, but should already be addressed in the results and discussions chapter:

*"(a) the ACE v.1 model in its current form cannot deal with fast changes in sedimentation rate (b) the signal to- noise-ratio of the LR04 **(age model [should be added])** is superior to the ratio of a single record and (c) no other a priori information **(in the ACEv.1 analysis contrary to the LR04 stack [should be added])** coming from other geological constraints is included as that the obliquity band should be in a certain frequency range*

**There is a lower signal amplitude at 41kyr at this time (Fig.3D), but this is also the case below 150m where the agreement with LR04 is rather good …**

This is an excellent observation. Between 60-90 m there is indeed a lower signal amplitude, as well as below 150 m. The difference is however that around 70-80m there is an elevated signal amplitude around the frequency of 0.5 cycle m$^{-1}$ (with the 41 kyr component mainly around 0.6 cycle m$^{-1}$) whereas there are no other elevated amplitude signals near the obliquity frequency below 150 m. This is also illustrated in a small drop (instead of rise) in sedimentation rate between 70-80 m in Figure 3G, potentially giving a hint that the lower boundary in the selected frequency range for the component estimation should be increased. This will be included in the discussion as well as the used frequency range for the component estimation (as is mentioned in the two other case studies).

**The LR04 sedimentation rate is rather flat, so there is (a priori) no strong change in the record at this time.**

The original LR04 sedimentation rate is not flat. It seems flat because we use a 500-point moving average. This is mentioned in the heading of the Figure but can be added for clarity purposes to the legend in the Figure. Additionally we can plot the non-averaged derived sedimentation rates too.

**The explanations given in the conclusion (page 13 lines 10-16) are therefore not fully convincing for this particular case study.**

We believe that with additional discussion and the rearrangement of our interpretation the analysis and its results should be clearer. However, the ACEv.1 model in its current form has its limitations. But these are also well illustrated with this specific case study. Therefore we believe it is and stays very valuable for the quality of the manuscript.

**3/ Clearly, the Danian record is the best example of the added value of the method (since there are little stratigraphic constraints except cyclostratigraphy). The advantages of the new method are discussed in the text (page 13, lines 17-25) but not well illustrated on the figures. It would be quite easy, and very helpful, to add on Figure 4F indications of alternative sedimentation rates (traditionnal tuning by Sinnesael et al 2016, …) somewhat equivalent to Figure 3G.**

We follow the helpful suggestion that adding the alternative sedimentation rates (Galeotti et al., 2015 and Sinnesael et al., 2016) into Figure 4F, would enhance the illustrative power of the Figure. In analogy with Figure 3G (and 3/ AR #1) we will take care to be consistent in the use of the appropriate format for the added lines.

*Next to the feedback of both reviewers we foresee the possibility to make esthetical changes in the provided MATLAB scripts, this based on user feedback. These would include a relaxation of the preconditions of the format of the input data and the implementation of a basic graphic output. Furthermore the equation numbering in the Appendix will be corrected.*

---

## Author Response (AR1)

Sinnesael et al. appreciate the review and suggestions for the discussion paper "Astronomical component estimation (ACEv.1) by time-variant sinusoidal modeling" by two anonymous referees.

In this document we provide a point-by-point answer to the interactive comments. These comments will be taken into account while preparing the final version of this manuscript.

5 **Anonymous Referee #1**

**\*Specific comments**

**1/ My main concern in the proposed approach is the lack of any measure of uncertainty associated with the estimation procedure. Error bars for the parameters estimated by linear least squares should be easily obtainable, and included in the derived instantaneous amplitude and frequency results.**

10 Providing a measure for uncertainty on the estimations is the main concern for both referees. We recognize that in the current version of the algorithm- uncertainties related to the sinusoidal fitting procedure are not taken into account. We want to meet this concern by discussing it in the methodology and by implementing a measure for the uncertainty in the ACEv.1 Matlab routines. We are currently exploring different techniques to estimate that uncertainty and to translate this uncertainty into sedimentation rate (and thus geologic time).

15 A first possibility to address this would be to use the uncertainties (standard deviations from the least-squares) off the parameters which are used to estimate the waveforms. However, the non-linear relationship between the parameters and instantaneous frequency (sedimentation rate), implies no closed-form solution for the uncertainties. For a single data record, the number of parameters to estimate = (Number of orbital components) x (Polynomial order + 1) x 2 x (Number of analysis frames). Combining all uncertainties of the parameters with error propagation and in our case of overlapping (analysis
20 frames) would be computationally non desirable.

An elegant and commonly used alternative approach would be the use of Monte Carlo simulations to evaluate the significance of the measure off fit.

**2/ Fig. 1 (b) - the role of the "frame" in the figure is not clear**

25 The graphics of Fig. 1b will be reevaluated, made more clear (specified as "analysis frame" as it is actually used in the text of the manuscript. Also the use of the words "analysis frame" and "window" will be checked for consistency throughout the whole manuscript.

**3/ Fig. 3 (G) - the width of the line representing 41 kyr is not consistent**

30 It would indeed make more sense to have the 41 kyr line thinner and thicken the line representing the averaged (500 point) sedimentation rates which were derived from the LR04 age model. We will change this for consistency purposes.

**\*Technical corrections**

Thank you, these will be corrected.

**Anonymous Referee #2**

5    **\*Specific comments**

**1/ As also mentionned by the first reviewer, I believe it is critical to provide and discuss the uncertainties associated with the fitting procedure, and try to translate them into error bars in the final age models or sedimentation rates.**

See the written answer for "1/ Anonymous Referee #1", same remark.

10    **2/ I am also a bit frustrated by the lack of discussion on the results obtained with the ODP846 record (basically Fig.3G) which is only presented by the sentence : " Except for a small difference between 70 and 100m the match is close". How large is the mismatch in terms of absolute age ? Where does this mismatch actually come from ? There is a lower signal amplitude at 41kyr at this time (Fig.3D), but this is also the case below 150m where the agreement with LR04 is rather good … The LR04 sedimentation rate is rather flat, so there is (a priori) no strong change in the**

15    **record at this time. The explanations given in the conclusion (page 13 lines 10-16) are therefore not fully convincing for this particular case study.**

We appreciate the remark that more discussion is needed for the technical aspects of this particular case study. Below, we break down the comment into pieces and provide a point-by-point answer.

**I am also a bit frustrated by the lack of discussion on the results obtained with the ODP846 record (basically Fig.3G)**

20    **which is only presented by the sentence : " Except for a small difference between 70 and 100m the match is close".**

The mismatch is also discussed in the conclusions, however it is indeed better to rearrange the text and provide more in-depth information in the discussion itself and less in the concluding chapter.  See more in the points below.

**How large is the mismatch in terms of absolute age ?**

This will depend on the position down the core. Sometimes the 41 kyr estimate derived sedimentation rates will be higher

25    than the LR04 sedimentation rates, sometimes lower. This also means both over-and underestimations can sometimes cancel each other out. We will include the difference in total duration of both approaches, as well as this remark, in the discussion of the manuscript.

**Where does this mismatch actually come from ?**

This is mentioned in the conclusions, but should already be addressed in the results and discussions chapter:

*"(a) the ACE v.1 model in its current form cannot deal with fast changes in sedimentation rate (b) the signal to- noise-ratio of the LR04 (**age model [should be added]**) is superior to the ratio of a single record and (c) no other a priori information (**in the ACEv.1 analysis contrary to the LR04 stack [should be added]**) coming from other geological constraints is included as that the obliquity band should be in a certain frequency range*

**There is a lower signal amplitude at 41kyr at this time (Fig.3D), but this is also the case below 150m where the agreement with LR04 is rather good …**

This is an excellent observation. Between 60-90 m there is indeed a lower signal amplitude, as well as below 150 m. The difference is however that around 70-80m there is an elevated signal amplitude around the frequency of 0.5 cycle m$^{-1}$ (with

10 the 41 kyr component mainly around 0.6 cycle m$^{-1}$) whereas there are no other elevated amplitude signals near the obliquity frequency below 150 m. This is also illustrated in a small drop (instead of rise) in sedimentation rate between 70-80 m in Figure 3G, potentially giving a hint that the lower boundary in the selected frequency range for the component estimation should be increased. This will be included in the discussion as well as the used frequency range for the component estimation (as is mentioned in the two other case studies).

15 **The LR04 sedimentation rate is rather flat, so there is (a priori) no strong change in the record at this time.**

The original LR04 sedimentation rate is not flat. It seems flat because we use a 500-point moving average. This is mentioned in the heading of the Figure but can be added for clarity purposes to the legend in the Figure. Additionally we can plot the non-averaged derived sedimentation rates too.

**The explanations given in the conclusion (page 13 lines 10-16) are therefore not fully convincing for this particular**
20 **case study.**

We believe that with additional discussion and the rearrangement of our interpretation the analysis and its results should be more clear. However, the ACEv.1 model in its current form has its limitations. But these are also well illustrated with this specific case study. Therefore we believe it is and stays very valuable for the quality of the manuscript.

25 **3/ Clearly, the Danian record is the best example of the added value of the method (since there are little stratigraphic constraints except cyclostratigraphy). The advantages of the new method are discussed in the text (page 13, lines 17-25) but not well illustrated on the figures. It would be quite easy, and very helpful, to add on Figure 4F indications of alternative sedimentation rates (traditionnal tuning by Sinnesael et al 2016, …) somewhat equivalent to Figure 3G.**

We follow the very helpful suggestion that adding the alternative sedimentation rates (Galeotti et al., 2015 and Sinnesael et
30 al., 2016) into Figure 4F, would enhance the illustrative power of the Figure. In analogy with Figure 3G (and 3/ AR #1) we will take care to be consistent in the use of the appropriate format for the added lines.

*Next to the feedback of both reviewers we foresee the possibility to make esthetical changes in the provided Matlab scripts, this based on user feedback. These would include a relaxation of the preconditions of the format of the input data and the*
35 *implementation of a basic graphic output. Furthermore the equation numbering in the Appendix will be corrected.*

Sinnesael et al. appreciate the review and suggestions for the discussion paper "Astronomical component estimation (ACEv.1) by time-variant sinusoidal modeling" by two anonymous referees.

In this document we provide a line-by-line overview of the changes in the manuscript. These were based on the Authors Response (25/07/2016) following the reviews.

**p.1, line 22:** replaced "*sliding window*" by "*overlapping analysis frames*", as a consequence of remark by reviewer #1 about the role of the "frame".

**p.1, line 27:** added sentence "*Uncertainty analyses of the model estimates are performed using Monte Carlo simulations.*", as both reviewers requested additional uncertainty analyses.

10 **p.2, line 18:** replaced "*data window*" by "*analysis frames (i.e. moving window approach)*", as a consequence of remark by reviewer #1 about the role of the "frame".

**p.2, lines 20-21:** added sentence "*In this study the term analysis frame is preferred over the term (moving window) to avoid confusion with the window (or tapering) function terminology which is often encountered in signal processing too.*", as a consequence of remark by reviewer #1 about the role of the "frame".

15 **p.2, line 22:** replaced "*window*" by "*analysis frame*", as a consequence of remark by reviewer #1 about the role of the "frame".

**p.3, lines 16-17:** added sentence "*Additionally, a measure for the uncertainty of the model estimates is given by means of Monte Carlo simulations.*", as both reviewers requested additional uncertainty analyses.

**p.5; line 30:** "*principle*" becomes "*principal*", spelling mistake pointed out by reviewer#1.

20 **p.6, line 22:** correct symbol notation in eq. (13), "$\hat{f}$" becomes "$\hat{f}_k$".

**p. 8-9:** additional paragraph in the methodology, as both reviewers requested additional uncertainty analyses.

"*2.4 Uncertainty Analysis*

[revised manuscript text omitted]

**p.10, line 6-8:** added sentence "*As we are dealing with a synthetic case-study aiming to introduce the mechanisms of the ACE v.1 model, we do not yet implement the uncertainty analysis.*", as a consequence of remark by reviewer #1 about the role of the "frame".

35 **p.11, line 6:** added "*the single*" to ~19 kyr periods, for clarification.

**p.11, line 16:** replaced "*window*" by "*analysis frame*", as a consequence of remark by reviewer #1 about the role of the "frame".

**p.11, line 30:** restructured citations "*Muller and MacDonald, 1997; Lisiecki, 2010*"

**p.12, line 8:** replaced "*analysis window*" by "*analysis frame*", as a consequence of remark by reviewer #1 about the role of the "*frame*".

**p. 11-12:** Reworked the discussion on the ODP846 case-study to meet the comments of reviewer#2 on this subject.

"*This simple operation results in abrupt changes in sedimentation rate which cannot be captured by the ACE v.1 model approach. These abrupt changes originate from the principle that the LR04 age model is based on a stack, which enhances the signal-to-noise ratio compared to an analysis on a single core. Moreover, it is tuned to an ice model that is driven by the 21 June insolation at 65°N. Therefore, in this study a 500-point running average is taken to smooth the results (Figure 3I). Only the long term averaged trends are captured by our ACE v.1 estimation approach and not the fine-scaled sedimentation rates of the original LR04 age model. This is because: (i) the ACE v.1 model in its current form cannot deal with fast changes in sedimentation rate (ii) the signal-to-noise-ratio of the stacked LR04 age model is superior to the ratio of a single record and (iii) no other a priori information as that the obliquity band should be in a certain frequency range is used in the ACE v.1 analysis, contrary to the LR04 age model where other geological constraints are included. In the described approach, a frequency range is selected, which with the user's a priori knowledge could be associated with the 41 kyr obliquity [0.47 – 0.71] cycle m$^{-1}$ and let the algorithms extract its waveform. The corresponding age model is then created by using the estimated instantaneous frequency changes and the association with the 41-kyr obliquity period.*

*Except for a small difference between 60-90 m the match between the results of the ACE v.1 model and the averaged LR04 age model is close. Remarkably, the interval has a pronounced elevated uncertainty on its estimation (Figures 3G and 3H). The origin of the mismatch is the lower signal amplitude in this interval. Note the elevated signal amplitude around 70-80 m around the frequency of 0.5 cycle m$^{-1}$ whereas the 41-kyr component is mainly around the 0.6 cycle m$^{-1}$ (Figure 3C). This feature makes that the ACE v.1 model suggests a small drop in sedimentation rate between 70-80 m (Fig. 3I), whereas the LR04 age model suggests a gradual rise in sedimentation rate in this interval. This mismatch could be reduced by increasing the lower boundary of the selected frequency range for the obliquity component estimation. A similar feature on a smaller scale is detectable around level 40-50 m (Figure 3G). Between 140 m and the bottom of the core, there is low power in the 41-kyr frequency range. However, in contrast to the 60-90m interval, there are no neighboring (in the frequency domain) elevated signal amplitudes. The loss of power in this frequency range at this position translates in an elevation of the uncertainty on the estimations for this interval (Figures 3G and 3H). In general, the comparison between the ACE v.1 modeled sedimentation rates and LR04 sedimentation rates yield satisfactory results, as stratigraphic intervals where a mismatch exists are red-flagged by increased uncertainty on the ACE v.1 model. The sign of the mismatch in sedimentation rates is not consistent throughout the core, which means that over- and underestimates of sedimentation rates cancel each other out. In this particular case study, there seems to be an overestimation in total duration (compared to the LR04 age model) of ~5 %, which is significant but not so bad, considering that we simply used the tracking of the obliquity related frequency. The periodogram of the tuned record, based on the 41 kyr derived SR, reveals a much cleaner signal (Figure 3J). Using the comparison between the SR derived in this novel approach and deduced from the LR04 age model for the ODP 846 record (Figure 3I), the conclusion is that again the algorithm performs well in capturing the main averaged trend.*"

**p.13, line 27:** added "*(Figure 4B)*", an extra reference to a figure for clarification.

**p.14, lines 3-5:** added sentence "*Notice an increase in uncertainty at this transition (Figures 4G and H). The final uncertainties on the SR in this case-study (one standard deviation) stay however very small, several order of magnitudes smaller than in the ODP846 case-study.*", as both reviewers requested additional uncertainty analyses.

**p.14, lines 15, 20:** added *"(Figure 4J)"*, an extra reference to a new figure plotting the sedimentation rates according to Galeotti et al., 2015 and Sinnesael et al., 2016, as suggested by reviewer#2.

**p.15, lines 7-8:** added sentence "*Basic uncertainty analysis is provided too in order to be able to assess the size and distribution of the model estimate uncertainties.*", as both reviewers requested additional uncertainty analyses.

**p.15, lines 24-26:** added sentence "*Moreover, the practical use of the uncertainty analysis on the size and distribution of the model estimates is well exemplified.*", as both reviewers requested additional uncertainty analyses.

**p.16, line 8:** "*tracking*", spelling error pointed out by reviewer#1.

**p.16, lines 16-20:** Updating code availability statement:

"*The ACE v.1 model is designed in MATLAB ® and all scripts are available in the Supplementary Materials. These include a main script that enables the user to load data, define model parameters, use the functions and produce basic graphical output. Also the separate scripts of the three functions "OrbitalComponentEstimation", "SedimentationRateEstimation" and "UncertaintyAnalysis" are included as well as an instructive manual "Manual_ACEv1_Model_Sinnesael_etal_2016_GMD.txt".* "

**p.16, line 24-25:** extra acknowledgement "*David De Vleeschouwer was funded through European Research Council (ERC) Consolidator Grant "Earthsequencing" (Grant Agreement No. 617462).*"

**p. 16-17, lines 5, 6, 8, 10, 12, 20 and 21:** correcting the numbers of the equations used in the Appendix.

**p.17, line 10:** correct symbol notions in eq. A.3, replace subscripts "$N$" by "$P$".

**p.17, line 21:** correct symbol notions in eq. A.5, replace symbol "$\theta$" by "$\hat{\theta}$" to indicate that the symbol represents estimation.

**p.21, line 7:** complement reference by giving page numbers "*115-137*".

**p.22, lines 13-15:** added Figure 3 description because of added uncertainty analyses as requested by both reviewers: "*(G) The uncertainties on the estimated instantaneous frequencies (1σ). (H) The uncertainties on the sedimentation rate estimates (1σ).*"

**p.22, line 15:** extra specification of the plotted sedimentation rates in Figure 3I, as suggested by reviewer#2: "*(I) Comparison between the ACE v.1 estimated sedimentation rate and the derived non-averaged and 500-point averaged sedimentation rates for ODP Site 846 according to the age model of the LR04 stack (Lisiecki and Raymo, 2005) and the estimated change in sedimentation rate based on the modeled obliquity component.*"

**p.22, lines 25-26:** added Figure 4 description because of added uncertainty analyses as requested by both reviewers: "*(G) The uncertainties on the estimated instantaneous frequencies (1σ). (H) The uncertainties on the sedimentation rate estimates (1σ).*"

**p.22, lines 27-29:** extra figure with plotted sedimentation rates in Figure 3J, as suggested by reviewer#2: *"(J) Comparison between the ACE v.1 estimated sedimentation rate and the derived sedimentation rates in the studies by Galeotti et al. (2015) and Sinnesael et al. (2016)."*

**p.23, Figure 1:** changed "*frame*" to "*analysis frame*" as suggested by reviewer#1.

**p.25, Figure 3:** Two additional subplots (Figures 3G and 3H) to show the uncertainty estimates (both reviewers) and the non-averaged LR04 sedimentation rate model was plotted extra on subfigure Figure 3I (reviewer#2).

**p.26, Figure 4:** Two additional subplots (Figures 4G and 4H) to show the uncertainty estimates (both reviewers) and an extra figure with plotted sedimentation rates (Galeotti et al., 2015 and Sinnesael et al., 2016) in Figure 3J, as suggested by reviewer#2.

[revised manuscript text omitted]